# Study of Excipients in Delayed Skin Reactions to mRNA Vaccines: Positive Delayed Intradermal Reactions to Polyethylene Glycol Provide New Insights for COVID-19 Arm

**DOI:** 10.3390/vaccines10122048

**Published:** 2022-11-30

**Authors:** David Pesqué, Ramon Maria Pujol, Orianna Marcantonio, Ainhoa Vidal-Navarro, José María Ramada, Alba Arderiu-Formentí, Agustí Albalat-Torres, Consol Serra, Ana María Giménez-Arnau

**Affiliations:** 1Department of Dermatology, Hospital del Mar—Institut d’Investigacions Mèdiques, Universitat Autònoma de Barcelona (UAB) and University Pompeu Fabra (UPF), 08003 Barcelona, Spain; 2Department of Occupational Health, Hospital del Mar—Institut Hospital del Mar d’Investigacions Mèdiques (IMIM), Universitat Pompeu Fabra, 08003 Barcelona, Spain; 3CIBER de Epidemiología y Salud Pública, 28029 Madrid, Spain; 4Department of Medical Pharmacy, Hospital del Mar, 08003 Barcelona, Spain

**Keywords:** COVID-19 vaccine, COVID arm, intradermal test, polyethylene glycol

## Abstract

Background: Skin local reactions to mRNA COVID-19 vaccines have been linked to the use of vaccine excipients. The aim of the study is to evaluate the role of skin testing excipients in delayed skin reactions due to mRNA COVID-19 vaccines. Methods: Skin testing among a group of healthcare workers with skin reactions due to mRNA vaccines was performed. Patch testing and intradermal testing (IDT) with polyethylene glycol (PEG)-400, PEG-2000, trometamol, and 1,2-dimyristoyl-sn-glycero-3-phosphocholine were performed. Healthcare workers without skin reactions to vaccines were used for skin testing as controls. Results: Thirty-one healthcare workers (from a total of 4315 vaccinated healthcare workers) experienced cutaneous adverse vaccine reactions. Skin testing was performed in sixteen of the healthcare workers (11 delayed large local reactions (DLLR) and 5 widespread reactions). Positive IDT for PEG-2000 1% in DLLR was seen in 10 (90.9%) patients, in comparison with one (16.6%) individual with a delayed widespread reaction. Delayed positive IDT reactions for PEG-2000 1% on day 2 were observed in three (27.3%) patients with DLLR. Patch testing of the excipients was negative. Among 10 controls, only one exhibited a transient positive IDT reaction to PEG-2000 1%. Conclusions: Immediate and delayed reactions to IDT are frequently detected in patients with DLLR. The observation of positive delayed intradermal reactions to PEG disclosed only in patients with DLLR reinforces a possible role of PEG in the development of these reactions. Skin testing of other excipients is of little importance in clinical practice.

## 1. Background

Evidence on the effectiveness of vaccines against coronavirus has brought new insights to the COVID-19 pandemic. In this setting, skin vaccine-linked adverse effects have also been evidenced [1,2]. COVID-19-vaccine-induced skin reactions frequently correspond to delayed large local reactions (DLLR) or “COVID arms”, which in medical literature have been mainly reported for mRNA COVID-19 vaccines [3,4,5,6,7]. DLLR present a clinical picture characterized by erythematous plaques, with pain and swelling, which, unlike local reactions to other types of vaccines, start a week or more after the first or second dose of vaccine [3,4,5,6,7].

The mechanisms lying behind these reactions remain poorly understood. Widespread skin reactions are believed to be due to the activation of inflammatory pathways triggered by mRNA vaccines, [8] whereas local site reactions have been attributed to some vaccine excipients: polyethylene glycol (PEG)-2000, distearoyl-sn-glicero-3-phosphocholine, and trometamol [9]. PEG has not been used before as a vaccine excipient [10]. Different approaches have tried to establish the role of the different excipients of mRNA COVID-19 vaccines in skin reactions without success, and to date, the role of skin testing of these excipients has been mostly performed in type I immediate reactions [11,12,13,14,15,16]. The possible role of mRNA vaccine excipients for delayed reactions has not been elucidated [17]. To date, there is no consensus or protocol on the study of vaccine excipients in delayed reactions to mRNA COVID-19 vaccines.

The aim of this investigation is to characterize cutaneous adverse vaccine reactions to mRNA COVID-19 vaccines and evaluate the role of skin testing excipients in delayed skin reactions.

## 2. Methods

### 2.1. Patients

A prospective unicentric study of a tertiary referral hospital was conducted with the approval from the Local Ethical Committee (number 2021/9900/I). The recruitment period lasted 4 months in 2021 (1 January to 1 May). A registry of vaccine-related side effects in healthcare professionals was created by the Occupational Health Department, Hospital del Mar, in collaboration with the Dermatology Department.

All healthcare workers of the hospital were asked to declare any skin reactions after vaccination, via the use of a questionnaire, that could be found inside the corporative intranet. When a skin reaction was detected, patients were given an appointment for a dermatology visit. Inclusion criteria were healthcare workers vaccinated against COVID-19 with an mRNA COVID-19 vaccine with any skin manifestation within 21 days after vaccination. Vaccines used in our hospital are Pfizer^®^ BNT162b2 vaccine (Pfizer BioNTech, Mainz, Germany) or Moderna^®^ mRNA-1273 vaccine (Moderna, Inc., Cambridge, MA, USA). Exclusion criteria were other explainable causes or triggers unrelated to COVID-19 vaccination.

### 2.2. Clinical Protocol

Following a systematized protocol, the following variables were collected: sex, age, working category, type of skin reaction(s), diameter of the lesion (in the case it was an only lesion), location, associated local symptoms, type of vaccine, reaction(s) after the first or second dose, timing and duration of reaction(s), treatments, past medical history, presence of allergies, and allergy type. Skin conditions were graded, including grade 1 (local reaction without symptoms), grade 2 (local reaction with systemic symptoms), grade 3 (widespread macular, papular, or vesicular reaction), and grade 4 (erythrodermic, exfoliative or bullous generalized reaction), following the grading classification suggested by Català et al. [2].

Local reactions were classified according to the timing of appearance after vaccination, as suggested by previous studies [1,2]. The first group were “immediate local reactions”, which occur within 96 h after vaccination, [1,2] and are not accompanied by immediate type I allergy. The second group included local reactions occurring more than 4 days after vaccination, called DLLR [1,2]. Widespread delayed reactions were classified according to their clinical aspect [1,2].

In the case the skin reaction was suggestive of an allergic type I reaction (appearance of acute urticaria within 4 h after vaccine or anaphylactic reaction), the patient was referred to the Allergy Department, and no complementary tests were performed. In the case the reaction had occurred after the first dose, a prophylactic scheme based on both anti-histamines and topical steroids was set. Patients were contacted to assess relapses after the second dose. All patients gave written informed consent to participate and explicit consent to publish iconography.

### 2.3. Skin Testing Protocol

All patients who matched inclusion criteria without evidence of type I hypersensitivity reactions were offered to be skin-tested. Previously, a written consent was signed. Another written consent was required for skin biopsies. The testing protocol to evaluate delayed reactions was based on patch tests and IDT, which were performed with the clinical evidence and guidelines available at the time of the study [11,14,15].

IDT was performed in accordance with established standards, with readings after 20 min and 2 h and on D2. Reactions were considered positive if the diameter of the wheal was more than 10 mm [18]. Controls were histamine (positive control) and serum (negative control). Excipients were obtained from Sigma-Aldrich (Saint Louis, MO, USA) in its pure form, and in-house prepared. PEG-400, PEG-2000, and trometamol were diluted with normal saline 0.9%.

PEG testing concentrations at 0.001%, 0.01%, 0.1% and 1% were decided in accordance with Klimek et al.’s and Wenande et al.’s recommendations [11,16]. A stepwise approach, starting with PEG-400, was used for PEG (CAS Number: 25322-68-3) of two low molecular weights (400 and 2000), based on literature recommendations [16]. The stepwise approach performed did not include PEG of molecular weight bigger than 2000 since PEG-2000 is the form present in vaccines, and the aim of this study was not to evaluate immediate allergy to PEG, as patients had no suspicion or history of immediate allergy to PEG. In the case any patient developed an immediate generalized skin or systemic reaction, no further tests were performed.

For trometamol (CAS Number: 77-86-1), testing concentrations (0.001%, 0.01%, 0.1% and 1%) were decided according to Scala et al. recommendations [19]. 1,2-distearoyl-sn-glicero-3-phosphocholine was not available, due to which the homologous-related molecule 1,2-dimyristoyl-sn-glycero-3-phosphocholine (CAS Number: 18194-24-6) was considered. However, for 1,2-dimyristoyl-sn-glycero-3-phosphocholine, no recommendations were found and the same concentrations were applied.

Patch tests were performed according to European standards. Readings were performed on D2, D4, and D7. PEG-400 was patch tested at 1%, at 5% and pure pet. with Finn Chambers (8 mm diameter, Smart Practice) on Scanpor tape (Norgesplaster, Vennesla, Norway) [20,21]. PEG-2000 was patch tested at 1%, 5% and 10% pet [20,21]. Trometamol was patch-tested, and 0.5% and 1% aq. [22] 1,2-dimyristoyl-sn-glycero-3-phosphocholine could not be patch-tested.

For a second time, healthcare workers without skin reaction to mRNA vaccines were offered to participate with the aim of studying the positive results disclosed in the healthcare workers group. These controls also signed an informed consent and accepted skin testing.

## 3. Results

### 3.1. Demographic Baseline

A total of 4315 healthcare workers were vaccinated during the aforementioned period, with women comprising 62.9% of the sample. A total of 3343 healthcare workers received at least one dose of Pfizer, with 3327 (99.5%) receiving both doses during this period. Furthermore, 972 received at least one dose of the Moderna vaccine, with 851 (87.6%) achieving the two-dose goal during this period. Thirty-one healthcare workers (0.72%) presented with skin reactions attributed to mRNA vaccines and were included, with a median age of 42 years old (IQR 52-30) and with men accounting for 12.9% of the patients.

### 3.2. Clinical Characterization of Skin Reactions Due to mRNA COVID-19 Vaccines

Twenty-three individuals (74.2%) had received the Moderna vaccine and eight had received the Pfizer vaccine. Twenty-three patients (74.2%) presented with local site reactions, with eighteen (78.3%) corresponding to DLLR. Local symptoms were seen in 26 (83.8%) patients. Nine skin reactions were observed after the second dose. Relapses with the second dose were seen in four patients. Local reactions were grade 2 (*n* = 19, 82.6%), even if some grade 1 reactions were seen (*n* = 4). All generalized reactions were grade 3. All patients were treated with conventional therapies (anti-histamines and/or topical steroids and/or paracetamol) and no vaccine discontinuation was needed. No immediate type I hypersensitivity reactions was observed during this period at our hospital.

In terms of local reactions, DLLR had a median onset time of approximately 8 days and a median duration of practically 5 days. Most patients (*n* = 13) presented the reaction after the first shot, and three relapsed with the second dose despite the prophylactic scheme. Immediate local reactions presented a median onset of approximately 2 days with a median duration of 3 days. All immediate local reactions occurred after the first dose, and only one relapse was seen. The majority of local reactions (DLLR and immediate local reactions) were due to Moderna, in sixteen (88.9%) and five individuals, respectively.

The demographic baseline and clinical features are detailed in Table 1.

### 3.3. Patch Testing Results

All patch tests performed were negative for all the excipients and concentrations in the 16 patients.

### 3.4. Intradermal Testing Results in Patients with Delayed Reactions

Sixteen patients accepted being skin tested, eleven patients who had presented DLLR and five with widespread reactions (two delayed urticariform rashes, one pityriasis rosea-like eruption, one morbilliform or maculo-papular reaction, and one psoriasiform eruption).

In the case of DLLR, skin testing showed positive reactions to PEG. IDT of PEG-400 at 1% showed positive reactions in four patients. All these reactions remained positive at 2h and even one on D2 for PEG-400 at 1%. In terms of PEG-2000 IDT, ten patients developed positive reactions at 1% at 20 min and 2 h, and even three of these presented long-standing reactions that persisted on D2. Results were similar for 0.1% and 0.01%. All patients that presented positivity against PEG-400 had also presented positivity for PEG-2000. Correlation of clinical findings and skin testing showed that two of the patients with delayed positive IDT on D2 to PEG-2000 corresponded to patients who had presented DLLR relapses with the second dose and the other presented DLLR only with the second vaccine. In addition, some of the biggest vertical and horizontal diameters corresponded to patients who presented with delayed positive reactions to PEG.

For other delayed widespread reactions, no cases of delayed positive reactions were seen, and two transient reactions were evidenced: one patient presented a positive reaction to IDT for PEG-400 and another for PEG-2000 at different concentrations.

Immediate transient reactions to trometamol and 1,2-dimyristoyl-sn-glycero-3-phosphocholine in isolated individuals were interpreted as possible irritative non-relevant reactions. No immediate generalized skin reactions nor systemic or anaphylactic reactions occurred during IDT of the excipients, including PEG.

Skin testing results are detailed in Table 2 and the clinical features of skin-tested patients are detailed in Table 3. The numerical order of patients from 1 to 16 between Table 2 and Table 3 is equal. Clinical pictures can be seen in Figure 1.

### 3.5. Skin Testing Health Care Workers without Skin Reactions

A total of 10 healthcare workers without skin reactions were skin-tested (patch testing and IDT to PEG). Those were mostly women (*n* = 8), with a median age of 43 years (IQR 51-36). Nine but one had previously been vaccinated with mRNA vaccines.

Skin testing revealed only one positive transient reaction to PEG in a 42-year-old woman when IDT of PEG-2000 at 1% was performed at 20min with negative results at 2 h and D2. No other positive reactions were evidenced.

### 3.6. Skin Biopsy of PEG Reactions

Biopsies of IDT-positive reactions to PEG-2000 on D2 were performed in two patients who consented to the test. Those depicted discrete papillary dermal edema and superficial perivascular lymphocytes with occasional eosinophils. No epidermal changes were noted.

## 4. Discussion

### 4.1. Skin Reactions Features

The number of healthcare workers with skin reactions attributed to mRNA vaccines during the period of study was low, in accordance with what was described in trials [23]. This possibly implies that these reactions are also an infrequent adverse effect in real-clinical practice. Cutaneous adverse vaccine reactions were more commonly reported due to the Moderna vaccine (74.2%), despite only being administered in 22.5% of healthcare workers, and DLLR was the most frequently reported skin finding, as observed in previous literature [2]. Most reactions were observed after the first dose, with only 17.4% of patients presenting a relapse after the second dose, which is much lower than reported by McMahon et al. 43% [1]. In terms of demographic features, reactions were reported mostly in women (87.1%), possibly due to possibly higher reactogenicity in this group [24].

### 4.2. Skin Testing Results for PEG

A total of 12 patients presented with positive reactions to PEG-400 and/or 2000, the majority of whom (*n* = 10) had experienced episode(s) of DLLR, but also some patients with delayed widespread reactions (*n* = 2) and one control. The high rate of positive results observed with IDT using PEG of low molecular weight at low concentration are not in accordance with previous skin testing procedures to study immediate PEG allergy [14,15,16]. In previous studies, the rates of prick testing positivity when studying PEG immediate allergy were low, particularly at low concentrations and low molecular weight [14]. However, since IDT is controverted in suspicion of immediate allergy to PEG, there are fewer data available on the nature of IDT to PEG in this group. IDT is considered to be more sensitive than skin prick testing, which can also lead to false-positive and irritative reactions if the substance is injected at high concentrations [25].

PEG-400 and PEG-2000 are causes of allergic contact urticaria and eczema [21,26]. In addition, urticarial reactions have been described when PEG-containing products have been rubbed in open application tests [16]. Findings in this study favor the idea that PEG can trigger both immediate and delayed skin reactions. Despite positive IDT having been described in patients with immediate allergy to PEG, [27] our group of health workers did not present with any immediate reaction, and therefore, immediate reactions observed in IDT do not correspond to an IgE-dependent mechanism. Immediate reactions have also been associated with other mechanisms (direct mast cell activation and complement activation), but one study has already suggested that these mechanisms may not be sufficient to understand these reactions [28]. It is important to note that only one control, presented with a positive transient reaction to PEG, suggesting that patients who have not experienced skin reactions to mRNA vaccines may be less prone to present with positive skin testing.

The increased presence of positive long-standing reactions to PEG in patients with DLLR shows that PEG has the ability to induce delayed skin inflammatory responses, which could mimic a delayed local reaction. Additionally, these results were not seen in healthcare workers without skin reactions nor widespread delayed reactions. Clinically, patients that presented DLLR to both doses also presented positive delayed IDT. Therefore, delayed positive IDT in patients with previous relapsing DLLR may raise awareness of PEG leading to delayed hypersensitivity.

The mechanisms behind DLLR remain poorly understood. It has been stated that DLLR could correspond to forms of Arthus reactions [4] or to T-cell mediated hypersensitivity/delayed hypersensitivity. To date, DLLR has not shown vasculitis lesions that could suggest an Arthus reaction, and delayed hypersensitivity reactions would be expected to occur sooner and would require previous sensitization [3,4,7]. Previous sensitization in some individuals could be possible due to its presence in many daily products, even if both immediate and delayed PEG allergy are considered to be globally rare [14,29]. Even if PEG is present in different medical products (laxatives, oncologic drugs, etc.), the occurrence of occupational exposure and sensitization has not been described in the literature before.

However, in the present study, some observations may indicate that these reactions do not correspond with typical delayed hypersensitivity. Patients with DLLR and positive IDT presented negative patch testing to PEG. In previous literature, patch testing has proved that PEG-2000 delayed allergy, even when these tests were performed in the setting of delayed hypersensitivity to topical drugs [26]. Bianchi et al. have also observed positive late readings of IDT for PEG-2000 in patients having experienced type I immediate reactions to mRNA vaccines and have suggested that such reactions could correspond to non-specific immunological reactions that show cellular immune protection against the vaccine [30]. In this previous work, all vaccinated patients presented positive IDT reactions at 24 h and had previously presented with immediate reactions. These results are not comparable to the results described in this work, since the vast majority of positive IDTs were seen in patients with DLLR and not in vaccinated controls or patients who had experienced delayed widespread reactions. All the points addressed highlight the need to establish the exact nature of delayed local reactions to PEG and the mechanisms triggering DLLR.

Finally, revaccination in patients experiencing skin reactions after the first dose proved to be safe and broadly well-tolerated. This is in accordance with the previously observed tolerability in patients with delayed reactions and even PEG-delayed hypersensitivity [1,2,31].

### 4.3. Limitations

This study has limitations. In this case, IDT was performed in order to achieve a better understanding of the delayed reactions observed in mRNA vaccine reactions in patients with no suspicion of immediate allergy to PEG, but the role of skin testing these excipients with different approaches has not presented conclusive results [30,32,33,34,35,36]. Skin testing delayed skin reactions with these excipients is not standardized, and its specificity, validity, and reproducibility remain unknown since the PEG used for the study is a chemical that has not been yet studied for specificity. Another important limitation to this study is that IgG, IgE for PEG-2000, and interferon-releasing assay stimulated with PEG-2000 were not performed. Future studies that include the use of interferon-releasing assays [37] would be of importance for the assessment of delayed reactions.

## 5. Conclusions

This study brings to light the low frequency of adverse skin effects to mRNA COVID-19 vaccines. During the follow-up of our cohort (4315 health workers), the occurrence of delayed reactions was very rare (<1%) and no immediate reactions were evidenced. The vast majority of delayed reactions corresponded to COVID arm reactions. Delayed reactions to mRNA COVID-19 vaccines were not a contraindication for subsequent vaccination since no severe generalized reactions were seen, even if relapses could occur. In terms of skin testing, the results of this study may suggest that skin-testing the other vaccine excipients (trometamol and 1,2-dimyristoyl-sn-glycero-3-phosphocholine) in patients with delayed reactions attributed to mRNA vaccines could not contribute significantly to the diagnostic and therapeutic management of these skin conditions, and it could be of limited clinical impact on real-clinical practice. Future studies will have to confirm this hypothesis. In terms of PEG testing, the rather frequent observation of both immediate and delayed reactions to IDT of PEG-400 and −2000 at low concentrations in individuals with DLLR raises the possibility of PEG triggering different types of skin reactions, including immediate reactions due to non-IgE mechanisms and delayed reactions after vaccination with mRNA COVID-19 vaccines. However, it is our belief that only in selected patients with local reactions to mRNA vaccines associated with particular features (e.g., previous history of delayed allergy to PEG-containing topical drugs, relapsing episodes of DLLR, exaggerated clinical pictures of DLLR) should the study of delayed PEG allergy be performed. Although delayed allergy to PEG continues to be rare, with an ever-increasing presence of PEG in a wide range of topical and injected medical products, and the suspicion of PEG as one of the triggers of mRNA cutaneous adverse vaccine reactions, the investigation of delayed hypersensitivity to PEG, in patients without suspicion of immediate allergy, may be seen as a challenge to tackle.

## Figures and Tables

**Figure 1 vaccines-10-02048-f001:**
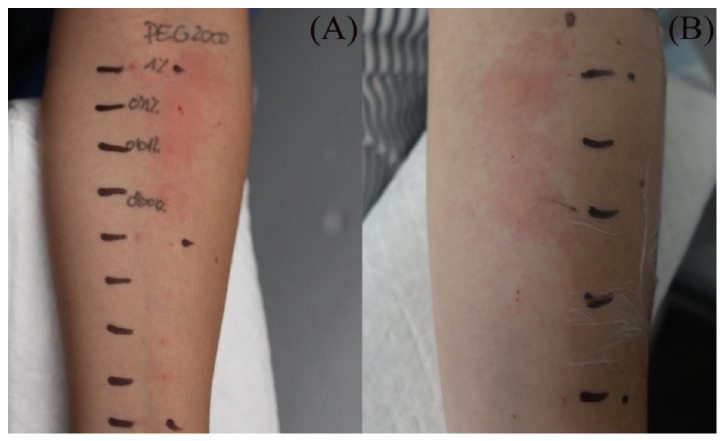
Skin testing: results for polyethylene glycol. (**A**) Intradermal testing (IDT) of PEG-2000 with positivity for all concentrations with significant papules and erythema at 20 min. (**B**) IDT of PEG-2000 with positivity for PEG-2000 1%, 0.1%, and 0.01% at 2 h.

**Table 1 vaccines-10-02048-t001:** Demographic baseline and clinical characterization of skin reactions due to mRNA COVID-19 vaccines.

Skin Reactions	DLLR	ILR	Morbilliform and Pityriasis Rosea-Like Rash	Urticariform Rash	Psoriasiform Rash	BMS
Clinical picture	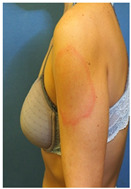 *n* = 18	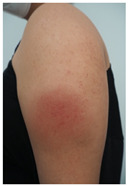 *n* = 5	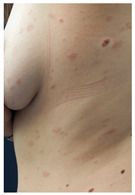 *n* = 4	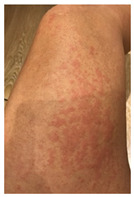 *n* = 2	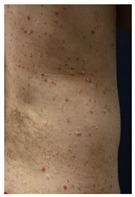 *n* = 1	No image *n* = 1
Median age (IQR)	38.9 (47–30)	47.6 (61.5–34)	43.8 (59.5–28)	35.5	72	44
Sex, *n* (%)						
Women	16 (88.9)	5	3	2	None	1
Men	2	None	1	None	1	None
Allergies, *n* (%)	11 (61.1)	1	2	1	1	1
Past anaphylaxis, *n*	2	None	None	None	None	None
Chronic skin disorder, *n*						
AD	3	2	None	1	None	None
CSU	3	1	1	1	None	None
Median onset (days) (IQR)	8.1 (9–7)	2.4 (3.5–1.5)	6.8 (8.5–5)	6	10	1
Median duration (days) (IQR)	4.8 (7–3)	3 (4.5–1.5)	4.3 (6–2.5)	6.5	15	10
Vaccine, *n* (%)						
Moderna	16 (88.9)	5	1	None	1	None
Pfizer	2	None	3	2	None	1
Dose, *n* (%)						
1st	13 (72.2)	5	3	None	None	1
2nd	5	None	1	2	1	None
Relapse with 2nd dose, *n*	3	1	None	None	None	None
Local symptoms, *n* (%)	16 (88.9)	3	4	2	None	1
Response to treatment	All patients presented a good response to conventional treatments					
Vaccine discontinuation	No vaccine discontinuation was needed for any patient with the 2nd dose					

Abbreviations: DLLR: delayed large local reactions; ILR: immediate local reactions; BMS: burning mouth syndrome; IQR: interquartile range; AD: atopic dermatitis; CSU: chronic spontaneous urticaria.

**Table 2 vaccines-10-02048-t002:** Skin testing delayed large local reactions (DLLR) and widespread skin reactions with intradermal tests.

**Skin** **Reaction**		**Intradermal Testing (Readings at 20 min/2 h/D2; if All Negative: Neg)**															
		**PEG-400**				**PEG-2000**				**TR**				**3-PC**			
		1.0%	0.1%	0.01%	0.001%	1.0%	0.1%	0.01%	0.001%	1.0%	0.1%	0.01%	0.001%	1.0%	0.1%	0.01%	0.001%
Widespread skin reactions	1	(+/+/−)	(+/+/−)	Neg	Neg	Neg	Neg	Neg	Neg	Neg	Neg	Neg	Neg	Neg	Neg	Neg	Neg
	2	Neg	Neg	Neg	Neg	Neg	Neg	Neg	Neg	Neg	Neg	Neg	Neg	Neg	Neg	Neg	Neg
	3	Neg	Neg	Neg	Neg	(+/+/−)	(+/+/−)	(+/−/−)	Neg	Neg	Neg	Neg	Neg	Neg	Neg	Neg	Neg
	4	Neg	Neg	Neg	Neg	Neg	Neg	Neg	Neg	Neg	Neg	Neg	Neg	Neg	Neg	Neg	Neg
	5	Neg	Neg	Neg	Neg	Neg	Neg	Neg	Neg	Neg	Neg	Neg	Neg	Neg	Neg	Neg	Neg
“COVID ARMS” or DLLR	6	Neg	Neg	Neg	Neg	Neg	Neg	Neg	Neg	Neg	Neg	Neg	Neg	Neg	Neg	Neg	Neg
	7	(+/+/+)	(+/+/−)	Neg	Neg	(+/+/+)	(+/+/−)	(+/+/−)	Neg	Neg	Neg	Neg	Neg	(+/−/−)	Neg	Neg	Neg
	8	Neg	Neg	Neg	Neg	(+/+/−)	Neg	Neg	Neg	Neg	Neg	Neg	Neg	Neg	Neg	Neg	Neg
	9	(+/+/−)	(+/+/−)	(+/+/−)	Neg	(+/+/+)	(+/+/+)	(+/+/−)	(+/+/−)	(+/−/−)	(+/−/−)	(+/−/−)	(+/−/−)	Neg	Neg	Neg	Neg
	10	(+/+/−)	Neg	Neg	Neg	(+/+/−)	(+/+/−)	(+/+/−)	(+/+/−)	Neg	Neg	Neg	Neg	Neg	Neg	Neg	Neg
	11	Neg	Neg	Neg	Neg	(+/+/−)	Neg	Neg	Neg	Neg	Neg	Neg	Neg	Neg	Neg	Neg	Neg
	12	Neg	Neg	Neg	Neg	(+/+/−)	(+/−/−)	(+/−/−)	Neg	Neg	Neg	Neg	Neg	Neg	Neg	Neg	Neg
	13	(+/+/−)	Neg	Neg	Neg	(+/+/−)	(+/+/−)	(+/+/−)	Neg	Neg	Neg	Neg	Neg	Neg	Neg	Neg	Neg
	14	Neg	Neg	Neg	Neg	(+/+/−)	(+/+/−)	(+/+/−)	Neg	Neg	Neg	Neg	Neg	Neg	Neg	Neg	Neg
	15	Neg	Neg	Neg	Neg	(+/+/−)	(+/+/−)	(+/+/−)	(+/+/−)	Neg	Neg	Neg	Neg	Neg	Neg	Neg	Neg
	16	Neg	Neg	Neg	Neg	(+/+/+)	(+/+/−)	Neg	Neg	Neg	Neg	Neg	Neg	Neg	Neg	Neg	Neg

Skin testing of DLLR versus widespread reactions. Abbreviations: (+): positive; (−): negative; PEG: polyethylene glycol; TR: trometamol; 3-PC: 1,2-dimyristoyl-sn-glycero-3-phosphocholine; D2: Day 2.

**Table 3 vaccines-10-02048-t003:** Clinical characteristics of skin-tested patients.

Skin Reaction		Type	Grading	Duration (days)	Vertical D (cm)	Horizontal D (cm)
Widespread skin reactions	1	Psoriasiform eruption	3	15	NA	NA
	2	Pityriasis rosea rash	3	6	NA	NA
	3	Morbilliform rash	3	4	NA	NA
	4	Urticariform rash	3	7	NA	NA
	5	Urticariform rash	3	6	NA	NA
“COVID ARMS” or DLLR	6	NA	1	6	6.5	8.0
	7	NA	2	3	11.0	6.1
	8	NA	2	7	8.4	5.3
	9	NA	2	15	14.5	7.4
	10	NA	2	7	8.5	6.8
	11	NA	1	6	5.8	4.5
	12	NA	2	4	8.5	7.2
	13	NA	2	7	9.5	7.0
	14	NA	2	6	8.8	7.5
	15	NA	2	7	8.0	6.5
	16	NA	1	3	9.5	5.0

Abbreviations: NA: not applicable; D: diameter.

## Data Availability

The data presented in this study are available on request from the corresponding author. The data are not publicly available due to privacy restrictions.

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
