# Peer review of "Study of Excipients in Delayed Skin Reactions to mRNA Vaccines: Positive Delayed Intradermal Reactions to Polyethylene Glycol Provide New Insights for COVID-19 Arm"

_vaccines, 2022, doi:10.3390/vaccines10122048_

Round 1

Reviewer 1 Report

The authors performed intradermal skin tests using PEG-400 or PEG-2000 in 11 subjects with delayed large local reactions (DLLR) and 5 with widespread reactions. Delayed positive reactions to PEG-2000 1% on day 2 were noted in 3 with DLLR. I have several comments.

1)        In the discussion, principally, positive skin test implies IgE-mediated type I allergy. They discussed Arthus phenomenon caused by previous sensitization against PEG materials. Patch test was negative in all. The author should examine basohilic activation test, IgG and IgE antibodies against PEG. These tests were reported by Warren CM et al Assessment of Allergic and Anaphylactic Reactions to mRNA COVID-19 Vaccines with Confirmatory Testing in a US Regional Health System. JAMA Network Open 2021;4(9): e2125524. doi:10.1001/jamanetworkopen.2021.25524

2)        What is the difference among DLLR, COVID ARM, and ILR?

As minor comments

1)        Widespread reactions were mentioned in the text. Are there any differences in the size of eruption among DLLR? In page 2, line 74-77, they graded from 1 to 4. This grading is not used in the text.

2)        In table 2, authors demonstrated the results of skin reactions in 16 cases. Skin test was negative in Case 6. Did this case have COVID arm or ILR? Clinical features, grading of the eruption size, and the duration of eruption should be mentioned for each case to know the relationship with the skin tests. In page 5 lines157-158, five individuals with widespread skin reactions had clinically different characteristics of rash. The author should provide the clinical feature of each case number. Cases 6-16, is there any relation between the long diameter of the rash and the persistence of positive skin tests?

3)        In page 7 line 216, 13patients presented with positive rations. In table 2, 12 were positive.

The paper has no additional constructive information.

Author Response

Answer to REVIEWER 1:

Thank you for your time and effort in reviewing this article. We are grateful for your kind and constructive comments that have clearly contributed to improving and reshaping the content and message of this work. Please, find in the forthcoming lines our point-by-point replies.

MAJOR COMMENTS

-POINT 1. In the discussion, principally, positive skin test implies IgE-mediated type I allergy. They discussed Arthus phenomenon caused by previous sensitization against PEG materials. Patch test was negative in all. The author should examine basophilic activation test, IgG and IgE antibodies against PEG. These tests were reported by Warren CM et al Assessment of Allergic and Anaphylactic Reactions to mRNA COVID-19 Vaccines with Confirmatory Testing in a US Regional Health System. JAMA Network Open 2021;4(9): e2125524. doi:10.1001/jamanetworkopen.2021.25524

-Comment: Thank you for this remark, which needs of a detailed explanation and reasoning of the methodology underlying this study.

Firstly, the article mentioned (Warren CM et al Assessment of Allergic and Anaphylactic Reactions to mRNA COVID-19 Vaccines with Confirmatory Testing in a US Regional Health System. JAMA Network Open 2021;4(9): e2125524. doi:10.1001/jamanetworkopen.2021.25524) was performed for patients with suspicion of immediate IgE-mediated allergy to vaccines (mostly PEG), presenting a wide range of reactions (even anaphylaxis) in the first 10 to 120min after vaccination. This was not the case of our study, which only included patients with delayed reactions. In fact, having any suspicion of immediate allergy to PEG (immediate allergy to topical products, PEGylated chemotherapies, etc) or presenting with an immediate reaction after vaccination was an exclusion criterion of the study. In terms of the performance of basophil activation test (BAT) and IgE and IgG antibodies against PEG, it can be argued that in the aforementioned article it was performed in a different setting (immediate allergy). Therefore, these tests were not performed since there was no suspicion of immediate allergy in our patients, and the studies are not comparable.

Furthermore, were we allowed to reason deeply on this subject, there are some nuances that need to be highlighted to justify the lack of performing these techniques. BAT assesses IgE cross-linking and is a more precise allergic readout than measuring the concentration of allergen-specific IgE, for immediate reaction that are IgE-mediated (Santos AF et al. Basophil activation test: Mechanisms and considerations for use in clinical trials and clinical practice. Allergy. 2021;76:2420-2432). Clinical application of BAT requires analytical validation, clinical validation, standardization of procedures and quality assurance to ensure reproducibility and reliability of results (Santos et al. Allergy. 2021; 76:2420-2432), which has not been performed for PEG to the best of our knowledge. Still, to date, the lowest sensitivity for BAT is seen in the study of immediate drug allergies (e.g. antibiotics) (Santos AF et al. Allergy 2021;76:2420-2432). It is not our view that positive skin test implies IgE-mediated type I allergy. What it is stated in the discussion is that these reactions do not seem to correspond to an IgE-dependent mechanism. While immediate positive reactions to intradermal tests are normally associated to IgE type I allergy, there are other non-IgE mechanisms that may explain these positive results (Lim XR et al. Evaluation of Patients with Vaccine Allergies Prior to mRNA-Based COVID-19 Vaccination. Vaccines (Basel). 2022;10:1025. and Barbaud A. Skin testing and patch testing in non-IgE-mediated drug allergy. Curr Allergy Asthma Rep. 2014;14:442). As stated in the article, our group had no suspicion of immediate allergy to PEG in this group of patients, as they did not present with any immediate reaction after vaccination, and therefore all the reactions seen should be explain by non-IgE mediated mechanisms. The fact that patch tests were negative, does not exclude the possibility of a delayed sensitivity. It is widely known that in high molecular weight molecules, proving contact dermatitis only by patch testing may be difficult as many times the molecule does not penetrate sufficiently to provoke a delayed reaction, and therefore other skin tests need to be performed. In fact, it is normally considered that patch test is the first approach to performed in non-IgE drug allergy but negative results do not exclude a type IV hypersensitivity due to the low sensitivity of these tests (Barbaud A. Skin testing and patch testing in non-IgE-mediated drug allergy. Curr Allergy Asthma Rep. 2014;14:442). In addition, a delayed reaction to intradermal test may be indicative of a delayed cellular reaction.

-POINT 2.  What is the difference among DLLR, COVID ARM, and ILR?

-Comment: Thank you for this important remark. DLLR and COVID arm are synonyms as stated in Page 1 Line 38. Then, as mention in the Methods section “Local reactions were classified according to the timing of appearance after vaccination, as suggested by previous studies.1,2 The first group were “immediate local reactions”, which occur within 96h after vaccination,1,2 and should not be accompanied by immediate type I allergy. The second group, included local reactions occurring more than 4 days after vaccination, called DLLR.1,2 Widespread delayed reactions were classified according to their clinical aspect.1,2”. Then, ILR are local reactions that are not due to immediate allergy but that occur with a different timing than DLLR or COVID ARM. This definition was not created by our group, but by McMahon et al. (McMahon DE et al. Cutaneous reactions reported after Moderna and Pfizer COVID-19 vaccination: A registry-based study of 414 cases. J Am Acad Dermatol. 2021;85:46-55) and Catala et al (Català A et al. Cutaneous reactions after SARS-CoV-2 vaccination: a cross-sectional Spanish nationwide study of 405 cases. Br J Dermatol. 2022;186:142-152)

MINOR COMMENTS:

-POINT 3. Widespread reactions were mentioned in the text. Are there any differences in the size of eruption among DLLR? In page 2, line 74-77, they graded from 1 to 4. This grading is not used in the text.

-Comment: Thank you for this remark. It is true that patients were classified in accordance to the same gradings that were used in the article of Catala et al (Català A et al. Cutaneous reactions after SARS-CoV-2 vaccination: a cross-sectional Spanish nationwide study of 405 cases. Br J Dermatol. 2022;186:142-152). Finally, those were not included due to the fact that results were pretty homogenous and similar to previous literature. Local reactions were grade 2 (n=19, 82.6%), even if some grade 1 reactions were seen (n=4). All generalized reactions were grade 3. However, this information has been added to the text.  

-POINT 4.  In table 2, authors demonstrated the results of skin reactions in 16 cases. Skin test was negative in Case 6. Did this case have COVID arm or ILR? Clinical features, grading of the eruption size, and the duration of eruption should be mentioned for each case to know the relationship with the skin tests. In page 5 lines157-158, five individuals with widespread skin reactions had clinically different characteristics of rash. The author should provide the clinical feature of each case number. Cases 6-16, is there any relation between the long diameter of the rash and the persistence of positive skin tests?

-Comment: Thank you for this important remark. All cases from 6-16 presented with DLLR or COVID-ARM. It is also our belief that it is important to include all their clinical characteristics in the table, but to ease the readability to the readers we have chosen to create a new table with the aforementioned clinical variables following the same order of patients, as in the skin testing results. This new table has been added to the text as Table 3. In addition, some of the patients who presented delayed reactions to PEG associated some of the biggest diameters.

Table 3. Clinical characteristics of skin tested patients

Skin reaction

Clinical features

Grading

Duration (days)

Vertical diameter

(cm)

Horizontal diameter

(cm)

Widespread skin reactions

1

Psoriasiform eruption

3

15

NA

NA

2

Pityriasis rosea rash

3

6

NA

NA

3

Morbilliform rash

3

4

NA

NA

4

Urticariform

rash

3

7

NA

NA

5

Urticariform

rash

3

6

NA

NA

“COVID ARMS”or DLLR

6

NA

1

6

6.5

8.0

7

NA

2

3

11.0

6.1

8

NA

2

7

8.4

5.3

9

NA

2

15

14.5

7.4

10

NA

2

7

8.5

6.8

11

NA

1

6

5.8

4.5

12

NA

2

4

8.5

7.2

13

NA

2

7

9.5

7.0

14

NA

2

6

8.8

7.5

15

NA

2

7

8.0

6.5

16

NA

1

3

9.5

5.0

Abbreviations: NA: not applicable

-POINT 5.  In page 7 line 216, 13patients presented with positive rations. In table 2, 12 were positive.

-Comment:  Thank you for this comment. This was a typographic error, as it is afterwards explained that positive results were seen in 10 patients with DLLR and 2 with delayed widespread reactions.  It has been changed to 12.

-FINAL REMARK.  The paper has no additional constructive information.

-Comment: We have reshaped the conclusion to make clear that this paper. Firstly, the follow-up of our prospective cohorts enhances that delayed reactions to mRNA COVID-19 vaccines are rare and immediate reactions are even rarer (not even one was evidenced in our cohort). To date, it is one of the biggest cohorts followed in the literature. Secondly, and in terms of skin testing, it has been revealed that intradermal tests with some important excipients (tromethamol, 3-PC) is of little benefit and it could be advised not to study these vaccine excipients in patients due to its low rentability.  However, the fact that PEG testing may be accompanied by delayed reactions enhances the possibility that its study in selected patients who might have clinical suspicion of PEG or repeatedly episodes of COVID ARM after vaccination is performed not only via patch testing but also with skin testing.

Therefore, we have summarized the most important findings in this new conclusion:

“This study brings into light the low frequency of vaccine skin adverse effects to mRNA COVID-19 vaccines. During the follow-up of our cohort (4315 health workers), the occurrence of delayed reactions was very rare (<1%) and no immediate reactions were evidenced. The vast majority of delayed reactions corresponded to COVID ARM reactions. Delayed reactions to mRNA COVID-19 vaccines were not a contraindication for subsequent vaccination, even if relapses could occur. In terms of skin testing, the results of this study indicate that systematic skin testing the other vaccine excipients in patients with delayed reactions attributed to mRNA vaccines does not contribute significantly to the diagnostic and therapeutic management of these skin conditions and is therefore not justified in real-clinical practice. In terms of PEG testing, the rather frequent observation of both immediate and delayed reactions to IDT of PEG-400 and -2000 at low concentration in individuals with DLLR, raises the possibility of PEG triggering different types of skin reactions, including immediate reactions due to non-IgE mechanisms and delayed reactions after vaccination with mRNA COVID-19 vaccines. However, it is our belief that only in selected patients with local reactions to mRNA vaccines that associate particular features (e.g. previous history on delayed allergy to PEG-containing topical drugs, relapsing episodes of DLLR, exaggerated clinical pictures of DLLR) the study of delayed PEG allergy should be performed. Despite delayed allergy to PEG continues to be rare, with an ever-increasing presence of PEG in a wide range of topical and injected medical products, and the suspicion of PEG as one of the triggers of mRNA cutaneous adverse vaccine reactions, the investigation of of delayed hypersensitivity to PEG, in patients without suspicion of immediate allergy, may be seen as a challenge to tackle.”

Reviewer 2 Report

This is potentially an important  pioneering study. However, as such, much more attention is needed. A positive recommendations can be given upon major revision carefully addressing the following Major Comments

Major Comment 1: Using Klimek et al 2020 as benchmarking PEG testing concentration should be carefully revisited. 

The problem is that Klimek et al 2020 was conjecture based article, no data were available at that time. Moreover, Klimek et al 2020 used skin test concentrations for systemically  administered  drugs based of sort of     an  ENDA/EAACI  Drug  Allergy  Interest Group  approach. However, as authors correctly mentioned in the limitations, specificity  validity and reproducibility remain unknown since the PEG used for the study is a chemical which has not been yet studied for specificity.  Therefore scenario B is requested to be analyzed, with lower thresholds for concentrations.

Major Comment 2: at page 3, line 122 authors written "The analysis included description of the data (chi^2-test for qualitative variables). P <0.05 was considered statistically significant. 

Please, state null and alternative hypotheses for this test. Are the prerequisites for application of chi^2-test verified?

Major Comment 3: in the section Discussion please provide clarification how p-values were computed. By chi^2 test? Pearson's chi-squared test is used to determine whether there is a statistically significant difference between the expected frequencies and the observed frequencies in one or more categories of a contingency table. Typically the null hypothesis is set up that there is no difference between  the groups. That means your p-values are showing significant differences, correct? Please,  clarify your null hypothesis and software used.

Major Comment 4: Please, write more in conclusions, these are too terse now. 

Author Response

Answer to REVIEWER 2:

Thank you for your time and effort in reviewing this article. We are grateful for your exhaustive comments and statistical review that has contributed to improving the understanding of the present manuscript. Please, find in the forthcoming lines our point-by-point replies.

MAJOR COMMENTS

-POINT 1. Using Klimek et al 2020 as benchmarking PEG testing concentration should be carefully revisited. The problem is that Klimek et al 2020 was conjecture based article, no data were available at that time. Moreover, Klimek et al 2020 used skin test concentrations for systemically administered  drugs based of sort of     an  ENDA/EAACI  Drug  Allergy  Interest Group  approach. However, as authors correctly mentioned in the limitations, specificity  validity and reproducibility remain unknown since the PEG used for the study is a chemical which has not been yet studied for specificity.  Therefore scenario B is requested to be analyzed, with lower thresholds for concentrations.

-Comment: Thank you for this important contribution. We agree with your comments. When the study was performed there was a lack in terms of knowledge of study of skin testing with vaccine excipients and our decision was based on the literature available at that moment. Note that the study began in January 2021. It is our firm believe those concentrations were correct as the risk of irritation with the concentrations used was low, but was significant enough to appreciate positive reactions.

-POINT 2.  At page 3, line 122 authors written "The analysis included description of the data (chi^2-test for qualitative variables). P <0.05 was considered statistically significant. Please, state null and alternative hypotheses for this test. Are the prerequisites for application of chi^2-test verified?

-Comment: Thank you for this important remark. Chi^2-test for qualitative variables prerequisites, which have been re-evaluated. In this study Chi^2 test was used to evaluate whether there were differences between the expected and the observed occurrence of reactions to vaccines for the category type of vaccines (Moderna and Pfizer). The same test was used to evaluate whether there were differences between the expected and the observed occurrence of reactions for the category sex (female or male). Therefore, the null alternatives were that the occurrence of vaccine reactions were equal for both sexes and for both types of vaccines.

In terms of revising the prerequisites, it is widely accepted that the usual rule for deciding whether the chi-squared approximation is good enough is that the chi-squared test is not suitable when the values in any of the cells of a contingency table are below 5. However, there is another more conservative rule that states that the test is not suitable when the values in any of the cells of a contingency table are below 10 when there is only one degree of freedom. In our case, the value for men with vaccine reactions was 4 and reactions due to Pfizer vaccine presented a value of 8. Therefore, it could be considered that the better approach to calculate it is the Fisher exact test. The calculation of Fisher exact test has been repeated for both this situation and, again, significancy has been found for differences in terms of occurrence of vaccine reaction and sex (p=0.0075) and occurrence of vaccine reaction and type of vaccine (p=0.00001). This has been changed in the text.

-POINT 3. in the section Discussion please provide clarification how p-values were computed. By chi^2 test? Pearson's chi-squared test is used to determine whether there is a statistically significant difference between the expected frequencies and the observed frequencies in one or more categories of a contingency table. Typically, the null hypothesis is set up that there is no difference between the groups. That means your p-values are showing significant differences, correct? Please, clarify your null hypothesis and software used.

-Comment: Thank you for this comment. The hypotheses have been answered in the previous point. P<0.05 indicates statistical significancy. Stata 17 was used for these calculations. This information has been added to the manuscript.

-POINT 4.  Please, write more in conclusions, these are too terse now.

-Comment: Thank you for this remark. The conclusions have been widened and it is now highlighted the strongest points. We have reshaped the conclusion to make clear that this paper. Firstly, the follow-up of our prospective cohorts enhances that delayed reactions to mRNA COVID-19 vaccines are rare and immediate reactions are even rarer (not even one was evidenced in our cohort). To date, it is one of the biggest cohorts followed in the literature. Secondly, and in terms of skin testing, it has been revealed that intradermal tests with some important excipients (tromethamol, 3-PC) is of little benefit and it could be advised not to study these vaccine excipients in patients due to its low rentability.  However, the fact that PEG testing may be accompanied by delayed reactions enhances the possibility that its study in selected patients who might have clinical suspicion of PEG or repeatedly episodes of COVID ARM after vaccination is performed not only via patch testing but also with skin testing.

Therefore, please find in the text the new conclusions:

““This study brings into light the low frequency of vaccine skin adverse effects to mRNA COVID-19 vaccines. During the follow-up of our cohort (4315 health workers), the occurrence of delayed reactions was very rare (<1%) and no immediate reactions were evidenced. The vast majority of delayed reactions corresponded to COVID ARM reactions. Delayed reactions to mRNA COVID-19 vaccines were not a contraindication for subsequent vaccination, even if relapses could occur. In terms of skin testing, the results of this study indicate that systematic skin testing the other vaccine excipients in patients with delayed reactions attributed to mRNA vaccines does not contribute significantly to the diagnostic and therapeutic management of these skin conditions and is therefore not justified in real-clinical practice. In terms of PEG testing, the rather frequent observation of both immediate and delayed reactions to IDT of PEG-400 and -2000 at low concentration in individuals with DLLR, raises the possibility of PEG triggering different types of skin reactions, including immediate reactions due to non-IgE mechanisms and delayed reactions after vaccination with mRNA COVID-19 vaccines. However, it is our belief that only in selected patients with local reactions to mRNA vaccines that associate particular features (e.g. previous history on delayed allergy to PEG-containing topical drugs, relapsing episodes of DLLR, exaggerated clinical pictures of DLLR) the study of delayed PEG allergy should be performed. Despite delayed allergy to PEG continues to be rare, with an ever-increasing presence of PEG in a wide range of topical and injected medical products, and the suspicion of PEG as one of the triggers of mRNA cutaneous adverse vaccine reactions, the investigation of of delayed hypersensitivity to PEG, in patients without suspicion of immediate allergy, may be seen as a challenge to tackle.”

Reviewer 3 Report

This is an interesting paper on reactions to Covid-19 vaccines, which shows most prevalent reactions to a specific excipient.
This is a third revision and obviously no major corrections are needed.

I have very minor comments, which I think may improve the paper.

1. 31 were included: Do you mean that 31 had reactions and were included or that more than 31 had reactions, but only 31 participated? When mentioning % included in the study it may be more useful to write that X% had a reaction to the vaccine, and Y% of those with reactions were included.
2. Please explain why skin biopsy was taken from 2 individuals. Others did not consent or the number was calculated beforehand, if so how?
3. Some healthcare workers may have been exposed to vaccines, PEG or others as part of their occupation. Do you have data on whether those with positive reactions may have been exposed previously as part of their occupation? If no data on this, it may still be worth mentioning and maybe speculate on possible previous exposure. Or perhaps it is known that no healthcare workers were exposed to these?

Author Response

Answer to REVIEWER 3:

Thank you for your time and effort in reviewing this article, and for your kind comments

Minor comments:

POINT 1. 31 were included: Do you mean that 31 had reactions and were included or that more than 31 had reactions, but only 31 participated? When mentioning % included in the study it may be more useful to write that X% had a reaction to the vaccine, and Y% of those with reactions were included.

-Comment: Thank you for this interesting comment. 31 presented with skin reactions attributed to mRNA and were therefore included in the study. We have included these changes in the manuscript.

POINT 2. Please explain why skin biopsy was taken from 2 individuals. Others did not consent or the number was calculated beforehand, if so how?

-Comment: Thank you for this question. Only 2 patients consented having a biopsy performed. This has been modified in the text.

POINT 3. Some healthcare workers may have been exposed to vaccines, PEG or others as part of their occupation. Do you have data on whether those with positive reactions may have been exposed previously as part of their occupation? If no data on this, it may still be worth mentioning and maybe speculate on possible previous exposure. Or perhaps it is known that no healthcare workers were exposed to these?

-Comment: The presence of PEG in some hospital products is known (laxatives, some oncologic treatments, etc). However, the occupational exposure and occupation sensitization is not known to the moment, according to our literature search. Therefore, we have added this to the manuscript.

Round 2

Reviewer 1 Report

The authors revised the manuscript of intradermal skin tests using PEG-400 or PEG-2000 in 11 subjects with delayed large local reactions (DLLR) and 5 with widespread reactions. The addition of Table 3 of detailed information on clinical characteristics increases readability. Still, I have major comments.

From the clinical findings, the authors considered that the DLLR did not seem to relate to the IgE-dependent mechanism. They performed intradermal testing and patch tests. They discussed the mechanisms of DLLR could associate with Arthus reaction or with T-cell mediated reactions. Patch tests were negative because of poor sensitivity. Certainly, delayed large eruptions seemed to be associated with some allergic reactions against PEG 2000 and the past sensitization is essential. Most readers considered that the positive intradermal skin test is IgE-mediated allergy. I suggested to perform serological study to detect IgG and IgE antibodies and interferon releasing assay (IGRA) stimulated with PEG 2000 to detect the previous sensitization and T cell immune response. As they discussed, laboratory data will support their results of intradermal skin test.

Author Response

Dear Prof. Dr. Tripp,

Thank you very much for your answer and reviewer comments of 7th November 2022. We are gladly resubmitting the revised manuscript (ID Vaccines-1982778) entitled “Study of excipients in delayed skin reactions to mRNA vaccines: positive delayed intradermal reactions to polyethylene glycol provide new insights for COVID arm” for consideration for publication in the journal Vaccines. We have revised the article according to the reviewer comments and suggestions. In addition, references have been revised, and methods and results adjusted to improve the legibility and understandability. Please find our point-by-point replies and the reviewed manuscript attached, that we have uploaded as a version with changes.

This manuscript describes original work and is not under consideration by any other journal. All authors approved the manuscript and this submission. Moreover, all authors listed have contributed sufficiently to the project to be included as authors.

We hope that this manuscript in its present form will be considered interesting to your readers and thus acceptable for publication in the journal Vaccines.

Thank you in advance for your kindness and looking forward to hearing from you at your earliest convenience.

Yours sincerely,

Ana M Giménez-Arnau, MD, PhD

Answer to REVIEWER 1:

Thank you for your time and effort in reviewing this article. We are grateful for your kind and constructive interest in trying to contribute and reshape our work. Please, find in the forthcoming lines our point-by-point replies.

MAJOR COMMENTS

-POINT 1. From the clinical findings, the authors considered that the DLLR did not seem to relate to the IgE-dependent mechanism. They performed intradermal testing and patch tests. They discussed the mechanisms of DLLR could associate with Arthus reaction or with T-cell mediated reactions. Patch tests were negative because of poor sensitivity. Certainly, delayed large eruptions seemed to be associated with some allergic reactions against PEG 2000 and the past sensitization is essential. Most readers considered that the positive intradermal skin test is IgE-mediated allergy. I suggested to perform serological study to detect IgG and IgE antibodies and interferon releasing assay (IGRA) stimulated with PEG 2000 to detect the previous sensitization and T cell immune response. As they discussed, laboratory data will support their results of intradermal skin test.

-Comment: Thank you for this remark and for raising awareness on this issue for this study of delayed reactions to mRNA vaccines and skin testing.  In terms of the complementary tests, the study of IgG and IgE antibodies, for immediate allergy, and IGRA stimulated with PEG-2000, for delayed allergy, could be of high interest to perform in this group of patients. However, our reference laboratory does not possess the techniques required to measure IgG and IgE antibodies for PEG, as it has been described before (e.g., Mouri M et al. Serum polyethylene glycol-specific IgE and IgG in patients with hypersensitivity to COVID-19 mRNA vaccines. Allergol Int. 2022;71:512-519). Other limitations to performing these techniques are that no serum samples of the participants were kept. As we have not performed these techniques, these has been included as an important limitation to this study: “Another important limitation to this study is that IgG, IgE for PEG-2000, as well as interferon releasing assay stimulated with PEG-2000 were not performed. Future studies including these techniques would be of importance for the assessment of delayed reactions”.

Reviewer 2 Report

Unfortunately, authors failed to satisfy majority of requests and comments.

I am listing the most of the problems. I cannot recommend publication of this manuscript, since in its present form is unfinished and biased.

MAJOR COMMENT 1. Unfortunately, remake of your analysis with lower concentration should be imperatively carefully revisited.

In science, I cannot accept believes, like "It is our firm believe those concentrations ..." that is not rigorous. 

MAJOR COMMENT 2. Now null and alternatives are stated. However, prerequisites for chi^2  are not verified. 

MAJOR COMMENT 3. Authors should clarify what they mean by "Secondly, and in terms of skin testing, it has been revealed that intradermal tests with some important excipients (tromethamol, 3-PC) is of little benefit and it could be advised not to study these vaccine excipients in patients due to its low rentability. " What is the rentability here? Cost-benefit? There is no budget for it?

MAJOR COMMENT 4. Authors need to be better explain "Delayed reactions to mRNA COVID-19 vaccines were not a contraindication for subsequent vaccination, even if relapses could occur. " Why it is not a contradiction?  Some explanation is needed. 

Author Response

Dear Prof. Dr. Tripp,

Thank you very much for your answer and reviewer comments of 7th November 2022. We are gladly resubmitting the revised manuscript (ID Vaccines-1982778) entitled “Study of excipients in delayed skin reactions to mRNA vaccines: positive delayed intradermal reactions to polyethylene glycol provide new insights for COVID arm” for consideration for publication in the journal Vaccines. We have revised the article according to the reviewer comments and suggestions. In addition, references have been revised, and methods and results adjusted to improve the legibility and understandability. Please find our point-by-point replies and the reviewed manuscript attached, that we have uploaded as a version with changes.

This manuscript describes original work and is not under consideration by any other journal. All authors approved the manuscript and this submission. Moreover, all authors listed have contributed sufficiently to the project to be included as authors.

We hope that this manuscript in its present form will be considered interesting to your readers and thus acceptable for publication in the journal Vaccines.

Thank you in advance for your kindness and looking forward to hearing from you at your earliest convenience.

Yours sincerely,

Ana M Giménez-Arnau, MD, PhD

Answer to REVIEWER 2:

Thank you for your time and effort in reviewing this article. We are very sorry for the misunderstandings that may have occurred with previous comments that were sent, and it is our duty to present a new series of comments to further clarify this work.

MAJOR COMMENTS

-POINT 1. Unfortunately, remake of your analysis with lower concentration should be imperatively carefully revisited. In science, I cannot accept believes, like "It is our firm believe those concentrations ..." that is not rigorous.

-Comment: Indeed, we are very sorry for the misunderstanding in relation to this point. Therefore, it is our aim to explain not only why do we think that the concentrations proposed by Klimek could be considered scientifically adequate, but also why those were adopted to our study. We never aimed to suggest that the choice of the concentrations was not rigorous or biased.

Firstly, it has been mentioned as a limitation that the sensitivity, specificity and validity of intradermal tests with PEG are limited, as there are no standards for the study of delayed reactions with these vaccines’ excipients. Therefore, when the study started, the concentrations were chosen with the evidence encountered at that moment to organize the study, prepare the excipients and present the project to the ethical committee.

Secondly, the article presented by Klimek et al. was chosen as a model to follow, because it considers a wide range of concentrations, from very low concentrations to quite high concentrations, with a ten-fold increase, also in line with the evidence of Bruusgaard-Mouritsen MA, et al. J Allergy Clin Immunol. 2022;149:168-175. In the study of hypersensitivity with intradermal reactions for delayed drug reactions, IDT are performed at low concentrations (normally 1/10000 or 1/1000) and there can be re-test at higher concentrations, gradually increasing them. There are several articles that shed light on this issue. It is important to highlight that concentrations are not standardized for most drugs or excipients. Some articles to be enhanced are: Brockow K, et al. Skin test concentrations for systemically administered drugs - an ENDA/EAACI drug allergy interest group position paper. Allergy. 2013;68:702–12; Barbaud A et. Intradermal Tests With Drugs: An Approach to Standardization. Front Med (Lausanne). 2020;7:156; Barbaud A et al. Skin Testing Approaches for Immediate and Delayed Hypersensitivity Reactions. Immunol Allergy Clin North Am. 2022;42:307-322; Barbaud A et al. Skin tests in the work-up of cutaneous adverse drug reactions: A review and update. Contact Dermatitis. 2022;86:344-356. In addition, as stated by the American Academy of Allergy, Asthma and Immunology, “smaller amounts are difficult to prepare due challenges in measuring very small aliquots of the concentrate” (https://www.aaaai.org/allergist-resources/ask-the-expert/answers/2021/intradermal-dilution).

Then, for PEG, the literature evidence (Bruusgaard-Mouritsen MA, et al. J Allergy Clin Immunol. 2022;149:168-175) suggest to avoid very high concentration of PEG, which is why the highest limit was placed at 1%, and the lowest limit was set in line with Klimek et al., since it is in line with previous publications (Bruusgaard-Mouritsen MA, et al. J Allergy Clin Immunol. 2022;149:168-175) and daily practice.

-POINT 2.  Now null and alternatives are stated. However, prerequisites for chi^2  are not verified.

-Comment: Thank you for this comment. Again, we are sorry for the misunderstanding. In the previous answer we did verify the prerequisites.

Please see in the previous answer were it was mentioned: “In terms of revising the prerequisites, it is widely accepted that the usual rule for deciding whether the chi-squared approximation is good enough is that the chi-squared test is not suitable when the values in any of the cells of a contingency table are below 5. However, there is another more conservative rule that states that the test is not suitable when the values in any of the cells of a contingency table are below 10 when there is only one degree of freedom. In our case, the value for men with vaccine reactions was 4 and reactions due to Pfizer vaccine presented a value of 8. Therefore, it could be considered that the better approach to calculate it is the Fisher exact test. The calculation of Fisher exact test has been repeated for both this situation and, again, significancy has been found for differences in terms of occurrence of vaccine reaction and sex (p=0.0075) and occurrence of vaccine reaction and type of vaccine (p=0.00001). This has been changed in the text.”

Therefore, in the previous answer, we revised the pre-requisites of the Chi-squared test (variables were categorical, independent, and cells in the contingency table mutually intelligible), but since in one case (men with vaccine reactions), the pre-requisites were not fulfilled (for sex and type of vaccine reactions only 75% of the cells had values greater than 5) we decided to repeat the calculation with the Fisher exact test, which, again, showed significancy. The current manuscript only includes the calculation with the Fisher exact test.

We thank you for this suggestion, and we will kindly await any further statistical recommendations if something would still not be clear.

-POINT 3. Authors should clarify what they mean by "Secondly, and in terms of skin testing, it has been revealed that intradermal tests with some important excipients (tromethamol, 3-PC) is of little benefit and it could be advised not to study these vaccine excipients in patients due to its low rentability. " What is the rentability here? Cost-benefit? There is no budget for it?

-Comment: Thank you for this comment. This is a language imprecision, for which we are very sorry, and we thank the referee for his/her accurate revision and improving our manuscript. The word “rentability” here can mislead to mistake, and has no relation to cost-benefit. The meaning of the sentence is that in our study no particular results were obtained through testing these excipients. Therefore, it could be hypothesized that the systematic performance of mRNA vaccines excipients (particularly, tromethamol and 3-PC) may not be justified. Therefore, we have enhanced the following sentence in the conclusions, which fits better with this idea: “In terms of skin testing, the results of this study indicate that systematic skin testing the other vaccine excipients (trometamol and 1,2-dimyristoyl-sn-glycero-3-phosphocholine) in patients with delayed reactions attributed to mRNA vaccines does not contribute significantly to the diagnostic and therapeutic management of these skin conditions and it could be of limited clinical impact on real-clinical practice.”

-POINT 4.  Authors need to be better explain "Delayed reactions to mRNA COVID-19 vaccines were not a contraindication for subsequent vaccination, even if relapses could occur. " Why it is not a contradiction?  Some explanation is needed.

-Comment: Thank you for this contribution. To date, the contradiction for vaccination are immediate reactions with an anaphylactic reaction or a proved immediate type I allergy. Another important contraindication is the presence of a severe generalized eruptions after vaccination (e.g, Stevens-Johnson syndrome) according to Dr. McMahon et al (McMahon DE, et al. Clinical and pathologic correlation of cutaneous COVID-19 vaccine reactions including V-REPP: A registry-based study. J Am Acad Dermatol. 2022;86:113-121). Since none of those were evidenced, no contraindications were seen. Nevertheless, some relapses were seen. Therefore, the sentence has been changed to: “Delayed reactions to mRNA COVID-19 vaccines were not a contraindication for subsequent vaccination since no severe generalized reactions were seen, even if relapses could occur”

Round 3

Reviewer 1 Report

According to their comments, there is no remaining samples and no laboratory methods.

It is too bad. Well, COVID-19 immunization with mRNA vaccines is still going on. Set up these laboratory examinations and please collect these DLLR cases. mRNA vaccine platform will be employed for influenza and other infectious disease or cancer therapy, not only for COVD-19. Therefore, it should be confirmed with laboratory tests.

Author Response

Answer to REVIEWER 1:

Thank you for your time and effort in reviewing this article.

MAJOR COMMENTS

-POINT 1. “According to their comments, there is no remaining samples and no laboratory methods. It is too bad. Well, COVID-19 immunization with mRNA vaccines is still going on. Set up these laboratory examinations and please collect these DLLR cases. mRNA vaccine platform will be employed for influenza and other infectious disease or cancer therapy, not only for COVD-19. Therefore, it should be confirmed with laboratory tests”

-Comment: We are thankful for suggesting the interferon-gamma release assay in evaluation of drug reactions, that we have read with great interest in previous publications (i.e. Kaplan et al. Fundam Clin Pharmacol. 2022;36:414-420). This technique is possible to be performed at our hospital, as it has already been confirmed by our laboratory, even if it is not validated for PEG, which may lead to increased false results. However, the technique is not implemented now in our laboratory and it would need to be implemented, inform the ethical committee, and look for patients willing to participate again and give blood samples. It would be a whole new study. We think that future studies could evaluate this, since our study already raises this hypothesis taking into account our clinical results. We have modified our limitations taking into account this point.

Reviewer 2 Report

Unfortunately, Authors did not made requested work and I cannot recommend publication of the manuscript, since it contains biased and not-scientific methods and approaches. Here follows major points to be done. Only then recommendation can be given.

Major comment 1: recalling my point 1: Unfortunately, adding lower concentrations is imperative, and avoiding doing so is not  scientific. All description in the reply does not replace direct experimentation. I cannot recommend acceptance of the article without doing such experiments.

Major comment 2: recalling my point 2: Categorical variables considered by authors are not independent. These are dependent. 

 Major comment 3: recalling my point 3: Authors need to reveal and explain the scientific backgrounds which are used in order to  hypothesized that the systematic performance of mRNA vaccines excipients (particularly, tromethamol and 3-PC) may not be justified. 

Author Response

Answer to REVIEWER 2:

Thank you for your time and effort in reviewing this article.

MAJOR COMMENTS

-POINT 1. Unfortunately, adding lower concentrations is imperative, and avoiding doing so is not scientific. All description in the reply does not replace direct experimentation. I cannot recommend acceptance of the article without doing such experiments.

-Comment: Thank you for your comment. This week we have contacted the participants of the study, and we have repeated the study with the excipients prepared by the Pharmacy Department in three patients (patients 8, 10 and 15) who accepted being re-tested. Even if they had signed the informed consent previously, it was signed again in sign of agreement. We have performed the study of the excipient at 0.0001%, which is a ten-fold dilution of the minimal concentration that was used at our protocol. All tests have been negative in the three patients.

-POINT 2.  Categorical variables considered by authors are not independent. These are dependent.

-Comment: Thank you for your comment. In order to avoid any confusion related to the relationship among variables, we have deleted the statistical comparisons between demographics and type of vaccines with the reactions’ incidence. Data is presented now only in a descriptive manner.

-POINT 3. Authors need to reveal and explain the scientific backgrounds which are used in order to hypothesize that the systematic performance of mRNA vaccines excipients (particularly, tromethamol and 3-PC) may not be justified.

-Comment: Thank you for this comment. This conclusion has been modified to be more hypothetical. The rationale used to suggest this is due to the results seen in our study, which, however, cannot be generalized. Thus, future studies will be needed to confirm this. 

The text has been changed to: “In terms of skin testing, the results of this study may suggest that systematic skin testing the other vaccine excipients (trometamol and 1,2-dimyristoyl-sn-glycero-3-phosphocholine) in patients with delayed reactions at-tributed to mRNA vaccines could not contribute significantly to the diagnostic and therapeutic management of these skin conditions and it could be of limited clinical impact on real-clinical practice. Future studies will have to confirm this hypothesis.”